# Male-killing symbiont damages host's dosage-compensated sex chromosome to induce embryonic apoptosis

Toshiyuki Harumoto[1,2], Hisashi Anbutsu[1], Bruno Lemaitre[2] & Takema Fukatsu[1]

Some symbiotic bacteria are capable of interfering with host reproduction in selfish ways. How such bacteria can manipulate host's sex-related mechanisms is of fundamental interest encompassing cell, developmental and evolutionary biology. Here, we uncover the molecular and cellular mechanisms underlying *Spiroplasma*-induced embryonic male lethality in *Drosophila melanogaster*. Transcriptomic analysis reveals that many genes related to DNA damage and apoptosis are up-regulated specifically in infected male embryos. Detailed genetic and cytological analyses demonstrate that male-killing *Spiroplasma* causes DNA damage on the male X chromosome interacting with the male-specific lethal (MSL) complex. The damaged male X chromosome exhibits a chromatin bridge during mitosis, and bridge breakage triggers sex-specific abnormal apoptosis via p53-dependent pathways. Notably, the MSL complex is not only necessary but also sufficient for this cytotoxic process. These results highlight symbiont's sophisticated strategy to target host's sex chromosome and recruit host's molecular cascades toward massive apoptosis in a sex-specific manner.

[1] Bioproduction Research Institute, National Institute of Advanced Industrial Science and Technology (AIST), Tsukuba 305-8566, Japan. [2] Global Health Institute, School of Life Sciences, École Polytechnique Fédérale de Lausanne (EPFL), Station 19, CH-1015 Lausanne, Switzerland. Correspondence and requests for materials should be addressed to T.H. (email: toshiyuki.harumoto@epfl.ch) or to T.F. (email:t-fukatsu@aist.go.jp).

The process, mechanism and origin of sex determination have been focal topics in genetics, cell biology, developmental biology and evolutionary biology[1–3]. Sex determination systems are strikingly diverse across animals, plants, fungi, protists and others, of which molecular mechanisms of sex determination and subsequent dosage compensation have been best documented for several model animals including fruit fly, nematode and mouse[3].

In the fruit fly *Drosophila melanogaster*, a female-specific developmental switch gene, *Sex lethal* (*Sxl*), counts autosome/sex chromosome ratio in an early developmental stage to establish the choice between male and female alternative developmental pathways at the cellular level. Downstream of *Sxl*, a cascade of regulatory genes branches into several major pathways, which respectively control sexual differentiation of the soma and neural cells, development of the germ line, and dosage compensation for equalizing X chromosomal transcript levels between males with XY chromosomes and females with XX chromosomes[4]. Dosage compensation is mediated by a ribonucleoprotein complex, designated as the male-specific lethal (MSL) complex, consisting of at least five proteins (MSL1, MSL2, MSL3, MLE (Maleless) and MOF (Males absent on the first)) and two non-coding RNAs (*roX1* and *roX2*), which concentrates on the single male X chromosome and up-regulates its transcriptional level approximately twofold[5].

Diverse insects and other animals, including *Drosophila* species, are commonly associated with symbiotic bacteria[6,7]. These microbial associates substantially influence their host's biology in a variety of ways. Some symbionts like *Wolbachia*, *Spiroplasma*, *Cardinium* and *Arsenophonus* cause striking reproductive phenotypes such as cytoplasmic incompatibility, male-killing, parthenogenesis and feminization, whereby these symbionts drive their own infection to spread into their host populations in selfish ways[8,9].

How these microbes interfere with host's reproduction and development is of fundamental interest, but the mechanisms have been poorly understood[9,10]. Previous studies provided some clues to the enigma in that symbionts are able to interact with a variety of eukaryotic molecular and cellular components including microtubules/centrosomes/mitotic spindles[11–16], paternal chromosomes[17–20] and somatic- and germline-stem cell niches[21,22]. As for male-killing *Spiroplasma* of *Drosophila* species, *msl* mutant hosts fail to express male-killing[23] and the infection alters the localization of the MSL complex[24], suggesting the involvement of the dosage compensation system, and infected male embryos suffer massive apoptosis[25,26] and neural malformation[26–28]. In *Ostrinia* moths, male-killing *Wolbachia* was reported to suppress host's masculinizing gene expression, thereby disturbing dosage compensation in male embryos[29]. In the light of these previous works, however, the processes as to how the symbiont's interactions with the host's molecular and cellular components are causally connected to host's reproductive phenotypes are elusive.

In this study, by making use of ample genetic tools and resources available for *Drosophila* in combination with sophisticated cytological, molecular and genomic techniques, we demonstrate a number of previously unrecognized molecular and cellular aspects of *Spiroplasma*-induced male-killing, which provide an integrative understanding of mechanisms underlying the symbiont's reproductive manipulation at the molecular, chromosomal, cellular and organismal levels.

## Results

**Transcriptomic analysis of infected and uninfected embryos.** We collected *Spiroplasma*-infected and uninfected *Drosophila* embryos of both sexes at stage 10–11 when infection-associated

male-specific abnormal apoptosis starts[26] (Fig. 1a). For embryonic sexing, we used a transgenic strain with green fluorescence protein (GFP) reporter of *Sxl* gene, *Sxl-Pe-EGFP*, which expresses GFP only in females (Fig. 1b and Supplementary Fig. 1a). Four groups of pooled embryos (uninfected females, uninfected males, infected females and infected males; three replicates for each group) were subjected to RNA-sequencing (RNA-seq) analysis. Of all the genes annotated in the *Drosophila* genome, 8,387 genes were substantially expressed in the embryos (Supplementary Methods), of which we identified 1,430 differentially expressed genes by all pairwise comparisons between the groups (false discovery rate < 0.001). Notably, more differentially expressed genes were associated with infected male embryos than other groups (Fig. 1c and Supplementary Fig. 1b). In infected male embryos, up-regulated genes were concentrated on the second and third chromosomes, whereas down-regulated genes were preferentially found on the X chromosome (Supplementary Fig. 1c). At a glance, this pattern may look like reflecting dosage compensation defects in infected male embryos. However, comparison with uninfected male embryos revealed only a small number of down-regulated X-encoded genes in infected male embryos (51 of 1,447 analyzed genes encoded on the X chromosome) (Supplementary Fig. 1c), suggesting that dosage compensation is still functioning in infected male embryos.

**Categorization of differentially expressed genes.** Of the 1,430 differentially expressed genes, 320 genes exhibiting at least twofold up- or down-regulation were selected and further analyzed. Hierarchical clustering grouped 314 genes into 6 clusters, whereas 6 genes were left ungrouped (Fig. 1d and Supplementary Data 1). Gene ontology (GO) analysis of these 6 clusters revealed that genes related to 'apoptosis' and 'DNA damage' were highly up-regulated in infected male embryos (cluster #1, 181/320 = 56.6%; Fig. 1e,f and Supplementary Data 1). These results are concordant with previous reports on the occurrence of abnormal apoptosis[25,26], and notably, indicative of a high level of DNA damage in infected male embryos. On the other hand, genes related to 'maintenance of gastrointestinal epithelium' were up-regulated in response to *Spiroplasma* infection irrespective of sex (cluster #2, 29/320 = 9.1%; Fig. 1e and Supplementary Data 1). Among them, *unpaired 1* (*upd1*) and *upd2* are ligands in the JAK-STAT (Janus kinase-signal transducers and activators of transcription) pathway, which are involved in host's survival upon intestinal-bacterial infection[30–32]. Furthermore, although not highlighted in the GO enrichment analysis, several genes related to 'detoxification', 'host defence' and 'stress response' were also identified (*GstE5*, *GstE9*, *Drsl5* and *proPO45*), which may reflect the general effects of *Spiroplasma* infection on host's physiology. Strikingly, no genes constituting the Toll and Imd (Immune deficiency) pathways were assigned to this cluster, which is in accordance with previous observations that *Spiroplasma* infection does not induce host's innate immune responses by evading host's recognition, presumably due to the absence of cell wall[33–35]. In male embryos irrespective of infection, 'dosage compensation' related genes were up-regulated, though small in number (cluster #3, 7/320 = 2.2%; Fig. 1e and Supplementary Data 1), confirming that embryonic sexing by the *Sxl-Pe-EGFP* transgene worked well (Supplementary Fig. 1a). *Spiroplasma* infection in male embryos were also associated with down-regulation of miscellaneous genes such as transcription, development, metabolism and so on (cluster #6, 50/320 = 15.6%; Fig. 1e and Supplementary Data 1), likely reflecting systemic attenuation of gene expression in the infected male embryos that exhibit developmental arrest leading to death. In the remaining

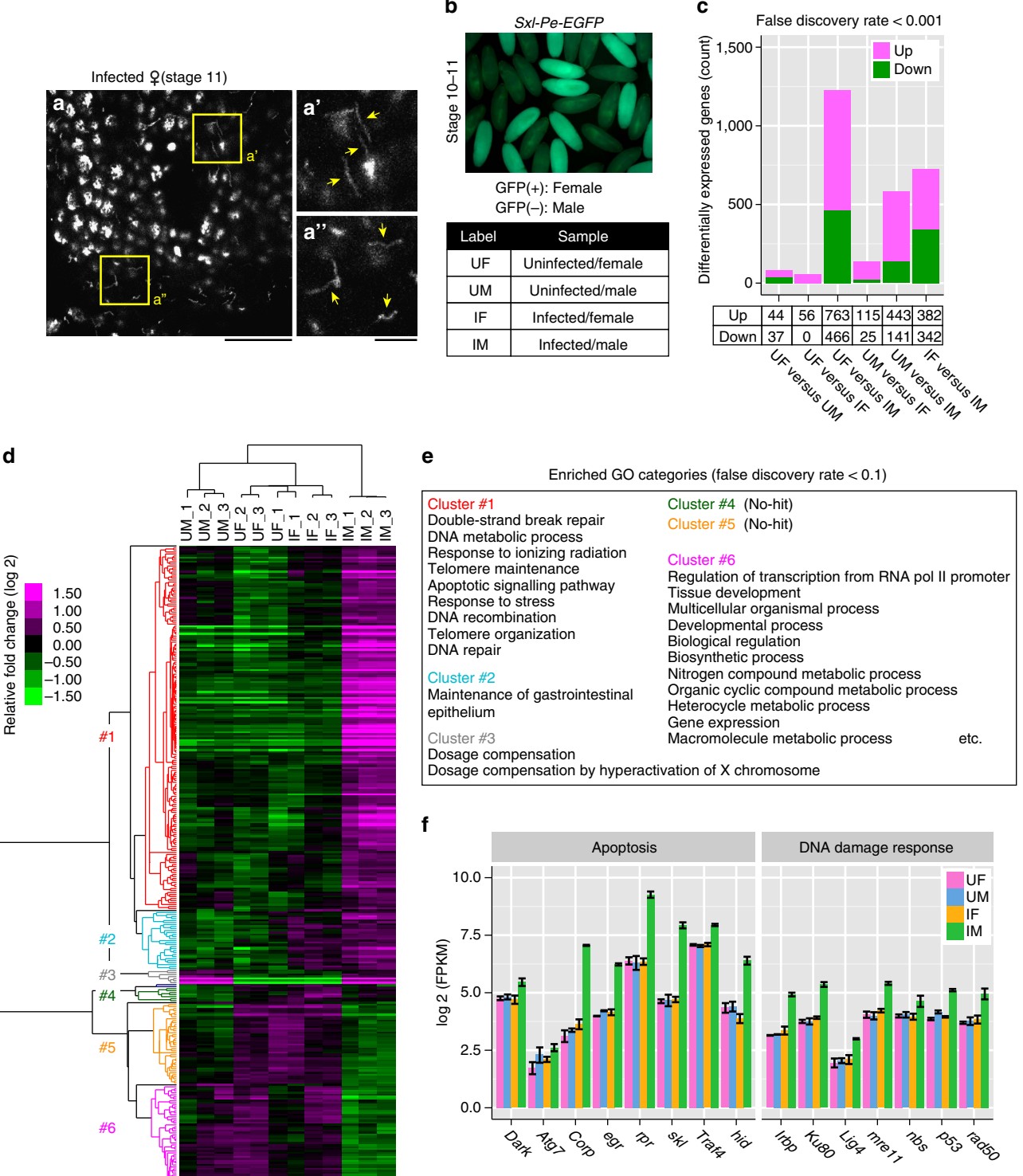

**Figure 1 | RNA-seq analysis of *Spiroplasma*-infected and uninfected embryos.** (**a**) Epithelial cells of an infected female embryo at stage 11. Three DNA-stained z-sections are projected to show the surface of the epithelium. Boxed regions in **a** are magnified in **a′** and **a′′** to highlight *Spiroplasma* cells (arrows). Scale bars, 20 μm (**a**) and 5 μm (**a′**,**a′′**). (**b**) Expression of *Sxl-Pe-EGFP* in uninfected embryos at stage 10–11. Only female embryos express GFP. Bottom panel shows abbreviations for RNA-seq samples. (**c**) The number of differentially expressed genes identified in pairwise comparisons between RNA-seq samples. In UF (uninfected female) versus UM (uninfected male), for example, numbers of differently expressed genes up- or down-regulated in UM compared with UF are plotted (the same applies hereafter). (**d**) A heat map of selected 320 differentially expressed genes. On the top is a clustering dendrogram of RNA-seq samples based on similarity of gene expression patterns across the samples. On the left is a clustering dendrogram of differentially expressed genes based on similarity of gene expression patterns across the genes, wherein gene clusters #1–#6 are depicted by different colours. (**e**) GO categories enriched in gene clusters #1–#6. (**f**) Expression levels (fragments per kilobase per million, FPKM) of major genes related to apoptosis (left) and DNA damage response (right) categorized to the gene cluster #1, represented as mean ± s.d. of three independent experiments.

two clusters (cluster #4, 7/320 = 2.2%; cluster #5, 40/320 = 12.5%), no GO terms were enriched and we could not find any distinctive features (Fig. 1e and Supplementary Data 1).

**DNA damage induces apoptosis in infected male embryos.** DNA damage is caused by a variety of genotoxic stresses including ionizing radiation, UV, chemicals, reactive oxygen species and replication stresses like stalling or delaying of replication fork progression. In response to DNA damage, a well-known tumour suppressor gene *p53* is activated, which triggers an assemblage of p53-dependent pathways to control cell cycle, DNA repair and apoptosis[36]. Our analysis using a fly strain with p53-responsive GFP reporter (*p53R-GFP*) revealed that p53 was strongly activated in infected male embryos (Fig. 2a,b). The *Drosophila* genome encodes a single *p53* family member, which is required for DNA damage-induced apoptosis[37–39]. Using a null allele of *p53*, we demonstrated that abnormal apoptosis in infected male embryos was significantly suppressed at stage 11–12 (Fig. 2c–f,i). On the other hand, developmental apoptosis prominent around the head region at these stages[26,40] was not affected (Fig. 2c,e,f, arrows). Upon severe DNA damage such as double-strand breaks caused by ionizing radiation, apoptosis is induced in a time-delayed manner even when p53 is absent[41,42]. Concordantly, p53-independent apoptosis was observed in infected male embryos from stage 14 onward (Fig. 2g,h,j). Taken together, these results strongly suggest that cells of *Spiroplasma*-infected male embryos suffer DNA damage, and then abnormal apoptosis is triggered via p53-dependent pathways.

**Differential detection of apoptosis and DNA damage.** Previous studies have established that, in response to DNA damage such as double-strand breaks and replication stress, a minor variant of histone H2A, called H2AX, is phosphorylated within the nucleus to form discrete foci, which are known as H2AX foci[43]. In *Drosophila*, an H2AX homologue, H2Av, has been reported to be phosphorylated following exposure to DNA damage[44] (Supplementary Fig. 2a,b). When *Spiroplasma*-infected male embryos were stained with an antibody against phosphorylated form of H2Av (pH2Av), two types of signals were detected: strong signals covering the whole nucleus (Supplementary Fig. 2d, yellow arrows) and relatively small bright foci located within the nucleus (Supplementary Fig. 2d, light blue arrowheads). In mammalian cells, detailed immunocytochemical studies on the distribution of phosphorylated form of H2AX (pH2AX) have demonstrated that strong nuclear-wide pH2AX signals are associated with apoptosis whereas intra-nuclear focal pH2AX signals represent DNA damage foci[45]. Our data suggest that these criteria also apply to pH2Av signals in infected male embryos as follows: (i) many, if not all, embryonic cells with strong nuclear-wide pH2Av signals were also apoptotic with TUNEL (terminal deoxynucleotidyl transferase dUTP nick end labelling) signals (Supplementary Fig. 2g, arrows); (ii) strong nuclear-wide signals were preferentially found in the head region, where developmental apoptosis occurs (Supplementary Fig. 2c,e,f, arrows); (iii) in control embryos, only fewer and relatively obscure focal signals were detected (compare Supplementary Fig. 2c,d); and (iv) when abnormal apoptosis was suppressed in infected male embryos mutant for *p53*, nuclear-wide signals were reduced while focal signals were still prominent (compare Supplementary Fig. 2d,f). In subsequent experiments, we focused on focal pH2Av signals as cytological indicators of DNA damage.

**DNA damage concentrated on the male X chromosome.** A previous study reported that maternal–zygotic *Drosophila* mutants for *msl* genes escape *Spiroplasma*-induced male-killing,

suggesting that dosage compensation of the single male X chromosome is required for male-killing expression[23]. Hence, we hypothesized that the male X chromosome bound by the MSL complex may be the target of *Spiroplasma*-induced DNA damage, and tested the hypothesis by visualizing the male X chromosome and DNA-damage foci simultaneously using anti-MSL1 and anti-pH2Av antibodies in the embryonic epidermal cells where *Spiroplasma*-induced abnormal apoptosis predominantly occurs[26]. In infected male embryos, MSL1 signals and pH2Av signals were frequently overlapping, while such overlapped signals were infrequent in control embryos (Fig. 3a–f). Quantitative analysis of the co-localized MSL1 and pH2Av signals revealed that significantly more focal pH2Av signals were located on the X chromosome of infected male embryos in comparison with control embryos (Fig. 3g,h), indicating that DNA damage is specifically enriched on the X chromosome of infected male embryos. These results support the hypothesis that the male X chromosome is a major target of *Spiroplasma*-induced DNA damage, which plausibly underlies the p53-dependent apoptosis observed in infected male embryos.

**Bridge breakage of the X chromosome in male embryos.** During the immunohistochemical experiments, we frequently observed inter-nuclear chromatin bridges in infected male embryos (Fig. 4d–f and Supplementary Fig. 3a). Notably, MSL1 signals frequently overlapped with chromatin bridges, suggesting that the male X chromosome may be involved in these abnormal structures (35/45 chromatin bridges observed in infected male embryos; Fig. 4e,f). To see more details, we analyzed 35 infected male embryos stained for both DNA and MSL1, and collected 140 mitotic cell images during anaphase, in which sister chromatids are about to separate and moving toward the opposite cell poles with a chromosomal bridge (Fig. 4g–i and Supplementary Fig. 3b–f). According to the extent of overlap between chromosomal bridges and MSL1 signals, we classified the images into three categories: only X, in which the chromosomal bridge and the MSL1 signal completely overlapped (116/140 = 83%; Fig. 4g–i and Supplementary Fig. 3c); with X, in which the chromosomal bridge and the MSL1 signal partially overlapped, or MSL1-labelled and unlabelled chromosomal bridges were observed simultaneously (21/140 = 15%; Supplementary Fig. 3d,e and Fig. 4i); and without X, in which the chromosomal bridge lacked the MSL1 signal (3/140 = 2%; Supplementary Fig. 3f and Fig. 4i). These results favour the idea that male X chromatids constitute chromosomal bridges. The chromosomal bridges were frequently associated with abnormally tangled DNA masses (95/140 = 68%; Fig. 4h and Supplementary Fig. 3c, arrows), suggesting compromised chromatin remodelling and/or condensation in male X chromatids. Notably, we observed that some X chromatids were asymmetrically segregated into two daughter cells (34/140 = 24%; Fig. 4h and Supplementary Fig. 3c,e, arrowheads), which presumably reflect the breakage of chromosomal bridges during cell division.

**The MSL complex is required for DNA damage and apoptosis.** While MSL1 and MSL2 act as scaffold for MSL complex formation, MSL3, MOF and MLE are required for spreading the complex across the entire X chromosome[5,46,47]. Loss-of-function mutants of *msl3*, for example *msl3¹*, fail to form the complete MSL complex and exhibit male-specific larval lethality due to dosage compensation defects[48–50]. On account of the maternal and zygotic sources of *msl3*, we investigated a maternal–zygotic mutant (*m−/z−*; zygotic genotype *msl3¹/msl3¹*) with compromised MSL complex function in comparison with a maternal mutant (*m−/z+*; zygotic genotype *msl3¹/TM3 ActGFP*) with

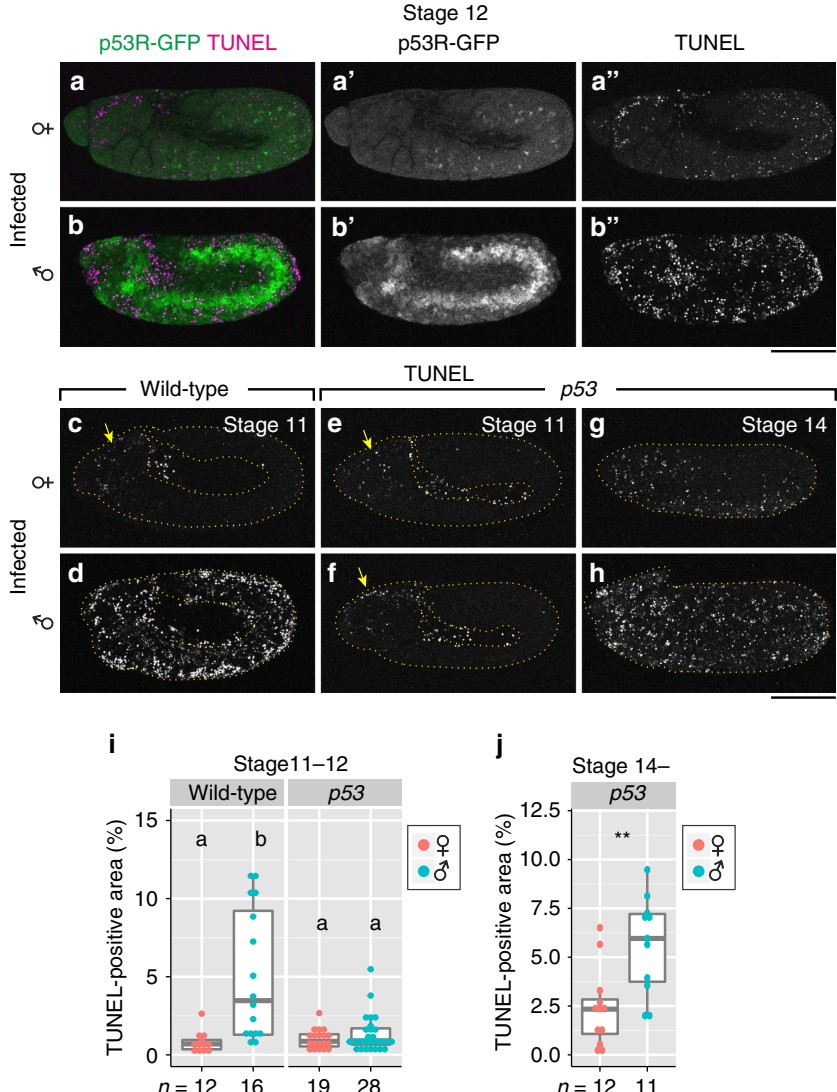

**Figure 2 | p53-dependent apoptosis in *Spiroplasma*-infected male embryos.** (**a**) *p53R-GFP* expression (green) and TUNEL staining (magenta) in an infected female embryo at stage 12 ($n = 12$). Single-channel images are shown in **a'** and **a''**. (**b**) An image similar to **a** of an infected male embryo ($n = 10$), wherein high p53 activity and massive apoptosis are seen. Single-channel images are shown in **b'** and **b''**. (**c**,**d**) TUNEL staining of infected female and male wild-type embryos at stage 11. The yellow arrow denotes developmental apoptosis in the head region. (**e**,**f**) Images similar to **c**,**d** of infected female and male embryos mutant for *p53*. (**g**,**h**) TUNEL staining of infected female and male embryos mutant for *p53* at stage 14. In **c–h**, the edges of embryonic epidermis are depicted by dashed yellow lines. (**i**) Quantification of TUNEL-positive areas in infected female and male embryos, wild type and mutant for *p53* at stage 11–12. Different letters (a,b) indicate statistically significant differences ($P < 0.01$; Kruskal–Wallis test followed by Mann–Whitney $U$-tests). (**j**) Quantification of TUNEL-positive areas in infected female and male embryos mutant for *p53* at stage 14 onward. Asterisks indicate a statistically significant difference (**, $P < 0.01$; Mann–Whitney $U$-test). In **i** and **j**, box plots indicate the median (bold line), the 25th and 75th percentiles (box edges), and the range (whiskers). Sample sizes are shown at the bottom. Scale bars, 100 μm.

the functional MSL complex. When these fly strains were infected with *Spiroplasma*, DNA damage and abnormal apoptosis in male embryos were attenuated under the *msl3*-deficient maternal–zygotic mutant genotype (Fig. 5a–c), indicating that the MSL complex is necessary for *Spiroplasma*-induced DNA damage and abnormal apoptosis.

**Ectopic MSL complex induces male-killing phenotypes.** In females of *Drosophila*, Sxl directly inhibits translation of MSL2 to prevent the formation of the functional MSL complex[5]. A previous study showed that the *H83M2* transgene, which encodes a suppression-resistant form of *msl2* mRNA, induces inappropriate dosage compensation of female X chromosomes

(Supplementary Fig. 4a–d), thereby causing reduced viability and developmental delay with a few escaper adult females[51] (Supplementary Fig. 4e, purple bars). When *H83M2* females were infected with *Spiroplasma,* no adult escapers were obtained (Supplementary Fig. 4f), suggesting occurrence of ectopic male-killing in infected females. It has been shown that *msl1* gene exhibits an allelic dosage effect in *H83M2* females, where even heterozygosity (*msl1/ +*) suppresses toxicity of this transgene due to reduced amounts of ectopic MSL complex[51]. We observed that both the deleterious effects and the ectopic male-killing were suppressed in *msl1^L60^/ +* heterozygotes (Supplementary Fig. 4e,f, red bars), supporting the notion that ectopic MSL complex formation is causative of these phenotypes. During the development of *Spiroplasma*-infected *H83M2* female embryos,

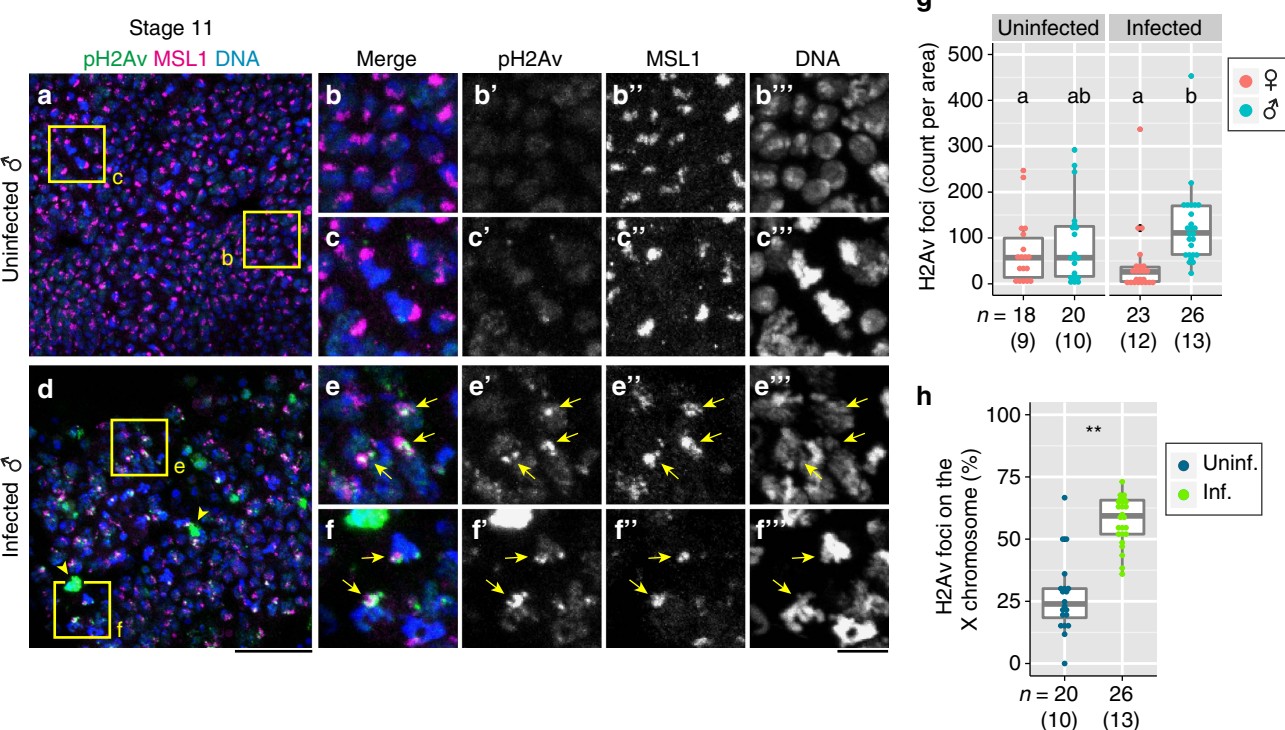

**Figure 3 | DNA damage in the X chromosome of *Spiroplasma*-infected male embryos.** (**a**) Simultaneous detection of pH2Av (DNA damage foci or apoptotic nuclei; green), MSL1 (X chromosomes; magenta) and DNA (blue) in an uninfected male embryo at stage 11, wherein few green pH2Av signals are seen. (**b,c**) Magnified images of boxed regions in **a**. Single-channelled images of **b** and **c** are shown in **b'–b'''** and **c'–c'''**, respectively. (**d**) An image similar to **a** of an infected male embryo, in which a number of green pH2Av signals are detected. (**e,f**) Magnified images of boxed regions in **d**. Single-channelled images of **e** and **f** are shown in **e'–e'''** and **f'–f'''**, respectively. In **d–f**, arrowheads indicate large pH2AX signals representing apoptotic nuclei, whereas arrows depict focal pH2AX signals representing DNA damage foci. (**g**) Quantification of focal pH2AX signals in uninfected and infected embryos at stage 11. Different letters (a,b) indicate statistically significant differences (*P* < 0.05; Kruskal–Wallis test followed by Mann–Whitney *U*-tests). (**h**) Quantification of focal pH2AX signals overlapping with MSL1-labelled X chromosomes in uninfected and infected male embryos at stage 11. Asterisks indicate a statistically significant difference (**, *P* < 0.01; Pearson's $\chi^2$ test). Focal pH2Av signals obtained in **g** were used to calculate the enrichment on the X chromosome in **h**. In **g** and **h**, box plots are as in Fig. 2i,j. Sample sizes (numbers of images analyzed) are shown at the bottom. Numbers of embryos inspected are shown in parentheses. Scale bars, 20 μm (**a,d**) and 5 μm (**b–c''',e–f'''**).

abnormal apoptosis was observed throughout the body (Fig. 5e,g) and chromatin bridges were frequently found (Fig. 5h–k). Without *Spiroplasma* infection, by contrast, *H83M2* females did not show these abnormal phenotypes (Fig. 5d,g,k), confirming that these abnormal phenotypes are associated with the *Spiroplasma* infection and are not ascribed to secondary effects of the ectopic MSL complex formation. Taken together, these results indicate that ectopic expression of the MSL complex can reproduce male-killing and associated cytological defects, including DNA damage, chromatin bridge formation and abnormal apoptosis in *Spiroplasma*-infected female embryos, which is in agreement with a recent report[24].

**Genetically dissecting effects of bridge breakage.** In an attempt to gain further insight into the relationship between DNA damage, chromosomal breakage and abnormal apoptosis, we genetically blocked cell division during embryogenesis. String (Stg), a CDC25 homologue of *Drosophila*, is essential for the initiation of G2/M transition in the cell cycle[52]. In zygotic mutants of strong alleles of *stg*, embryonic cells initially undergo normal cleavage cycles by using maternal transcripts during mitoses 1–13, and after cellularization when zygotically regulated cell division starts (from mitosis 14 onward), cells are arrested at G2 phase during the rest of embryogenesis, thereby resulting in embryos with fewer and larger cells[52] (Fig. 6a,b). Considering that the recruitment of the MSL complex to the male X chromosome

is first detected in cellularized embryos at mitosis 14 (refs 53,54), embryonic cells mutant for *stg* do not undergo cell division after the formation of the MSL complex, which is required for *Spiroplasma*-induced DNA damage. Therefore, using *stg* mutant embryos, we can genetically dissect whether bridge breakage in the male X chromosome has a causative role for the chromosome-specific DNA damage induced by *Spiroplasma*.

**Chromosome-specific DNA damage precedes bridge breakage.** In *Spiroplasma*-infected male embryos mutant for *stg*, abnormal apoptosis was significantly suppressed in comparison with control embryos (Fig. 6a–c), indicating that cell division is required for the expression of abnormal apoptosis. On the ground that chromosomal bridge-breakage occurs during mitosis, abnormal apoptosis is likely attributable to the DNA damage response activated by the bridge breakage in the X chromosome. On the other hand, even in *Spiroplasma*-infected male embryos mutant for *stg*, remaining apoptosis was observed around the head region (Fig. 6b), indicating that the blockage of cell division mainly suppresses p53-dependent abnormal apoptosis rather than developmental apoptosis. Despite the suppression of abnormal apoptosis, focal pH2Av signals on the male X chromosome were still prominent in *Spiroplasma*-infected male embryos mutant for *stg* (Fig. 6d,e), indicating that the male X chromosome has been damaged even in the absence of bridge breakage.

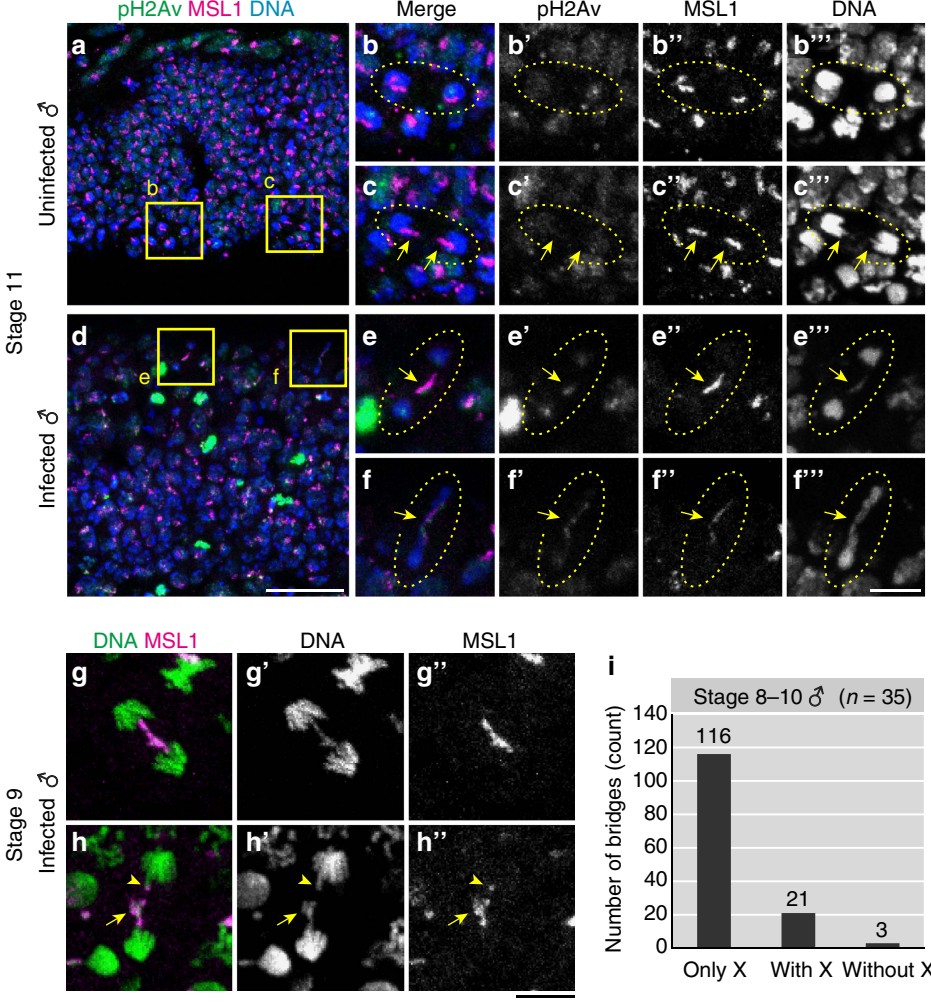

**Figure 4 | Bridge and breakage of the male X chromosome during mitosis.** (**a**) Epidermal cells of an uninfected male embryo at stage 11, in which pH2Av, MSL1 and DNA are visualized in green, magenta and blue as in Fig. 3a. (**b,c**) Magnified images of boxed regions in **a**, whose single-channelled images are shown in **b′–b′′′** and **c′–c′′′**, respectively. Dotted circles highlight dividing cells in telophase. In **b–b′′′**, sister chromatids are normally segregating to daughter cells, whereas in **c–c′′′**, MSL1-labelled X chromatids seem to be segregating slightly slower than the other chromatids (arrows). (**d**) An image similar to **a** of a *Spiroplasma*-infected male embryo, exhibiting many pH2Av signals. (**e,f**) Magnified images of boxed regions in **d**, whose single-channelled images are shown in **e′–e′′′** and **f′–f′′′**, respectively. Dotted circles highlight dividing cells in telophase, while arrows indicate inter-nuclear bridges overlapping with MSL1 and focal pH2Av signals, indicating that damaged male X chromatids constitute inter-nuclear bridges. (**g,h**) Two examples of anaphase chromatin bridges in infected male embryos at stage 9, wherein chromosomal DNA (green) and MSL1 representing X chromatids (magenta) are shown. Single-channelled images are shown in **g′, g′′** and **h′, h′′**. Arrows and arrowheads in **h–h′′** indicate an abnormally tangled DNA mass and asymmetrically segregated X chromatids, respectively. (**i**) Categorization of anaphase chromatin bridges in infected male embryos at stage 8–10. In total 140 anaphase bridges from 35 embryos were inspected. The categories 'only X', 'with X' and 'without X' indicate complete, partial and no overlap between chromatin bridges and MSL1 signals. For more detail, see text. Scale bars, 20 μm (**a,d**) and 5 μm (**b–c′′′**, **e–f′′′** and **g–h′′**).

**Partially attenuated neural disorder in *p53* mutant embryos**. In addition to abnormal apoptosis, disordered neurogenesis is among the most prominent defective phenotypes of *Spiroplasma*-infected male embryos; while highly organized central and peripheral nervous systems develop in control embryos, whole nervous systems are severely disorganized in *Spiroplasma*-infected male embryos[26–28] (Supplementary Fig. 5c,d). We examined whether and how DNA damage in the male X chromosome and subsequent activation of p53-dependent signalling pathways are relevant to *Spiroplasma*-induced neural defects. In neural precursor cells called neuroblasts, focal pH2Av signals were overlapping with MSL1 signals in *Spiroplasma*-infected male embryos (Supplementary Fig. 5a,b), indicating that DNA damage certainly occurs in the male X chromosome of neuroblasts as in the epidermal cells where p53-dependent abnormal apoptosis

occurs (Figs 2 and 3). When differentiated neural cells were visualized with an antibody against a specific marker protein Elav (embryonic lethal abnormal vision)[55], neural organization was severely disordered in *Spiroplasma*-infected control male embryos (Supplementary Fig. 5c,d), and notably, the neural disorder was partially recovered in *Spiroplasma*-infected male embryos mutant for *p53*: the overall morphology of the ventral nerve cord was restored considerably, but each neural cluster was still disorganized (Supplementary Fig. 5e,f).

**Apoptosis suppression similarly attenuates neural disorder.** Abnormal apoptosis in *Spiroplasma*-infected male embryos is concentrated on epidermal cells and scarcely associated with neural cells[26,28]. Considering that the massive apoptosis is

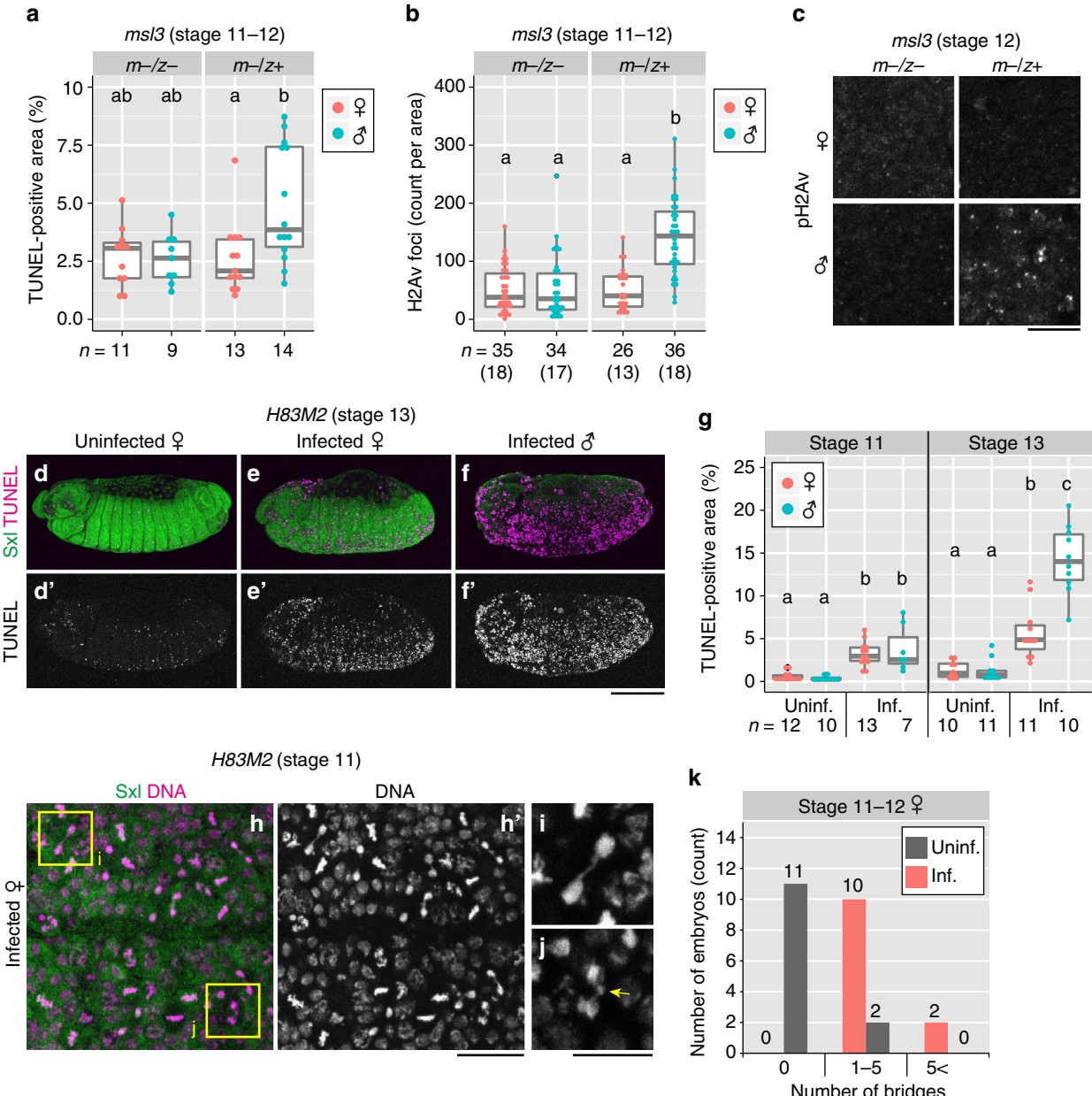

**Figure 5 | The MSL complex is necessary and sufficient for *Spiroplasma*-induced DNA damage and abnormal apoptosis.** (**a–c**) Apoptosis (**a**) and DNA damage (**b,c**) in *Spiroplasma*-infected male and female embryos of *msl3* maternal–zygotic mutant (*m − /z −* ; zygotic genotype *msl3^1^/msl3^1^*) and maternal mutant (*m − /z +* ; zygotic genotype *msl3^1^/TM3 ActGFP*). (**a**) Quantification of TUNEL-positive areas at stage 11-12. Different letters (a,b) indicate statistically significant differences (*P* < 0.05; Kruskal–Wallis test followed by Mann–Whitney *U*-tests). (**b**) Quantification of focal pH2Av signals at stage 11-12. Different letters (a,b) indicate statistically significant differences (*P* < 0.01; Kruskal–Wallis test followed by Mann–Whitney *U*-tests). (**c**) Focal pH2Av signals in *msl3* mutant embryos at stage 12. (**d-k**) Ectopic MSL complex formation by the *H83M2* transgene. (**d**) An uninfected *H83M2* female embryo exhibiting little abnormal apoptosis. (**e,f**) Infected *H83M2* embryos showing abnormal apoptosis (**e**, female; **f**, male). In **d-f**, stage 13 embryos are stained for Sxl (green) and TUNEL (magenta), whereas single-channelled TUNEL images are shown in **d'-f'**. (**g**) Quantification of TUNEL-positive areas in uninfected and infected *H83M2* embryos at stage 11 (left) and 13 (right). Different letters (a-c) indicate statistically significant differences (*P* < 0.01; Kruskal–Wallis test followed by Mann–Whitney *U*-tests). (**h**) Epidermal cells of an infected *H83M2* female embryo at stage 11, stained for Sxl (green) and DNA (magenta), whereas single-channelled DNA image is shown in **h'**. (**i,j**) Enlarged images of dividing cells with a chromatin bridge (**i**) and an abnormally tangled DNA mass (**j**, arrow), representing boxed regions in **h**. (**k**) Quantification of chromatin bridges in the epidermal cells of uninfected and infected *H83M2* female embryos at stage 11-12. The number of chromatin bridges per × 63 objective view are categorized into three classes: no bridge (0); 1 to 5 bridges (1-5); and 6 or more bridges (5<). In **a, b** and **g**, box plots are as in Fig. 2i,j. and sample sizes are indicated at the bottom. In **b**, numbers of embryos observed are shown in parentheses. Scale bars, 10 μm (**c, i** and **j**), 100 μm (**d-f'**) and 20 μm (**h,h'**).

suppressed in *p53* mutant embryos, the recovery of overall neural morphology may be attributable to suppression of the extensive cell death in surrounding non-neural tissues. To test this hypothesis, we analyzed the homozygous *H99* mutant in which

pro-apoptotic genes are deleted and apoptosis is almost completely blocked during embryogenesis[56]. When we examined *Spiroplasma*-infected male embryos deficient for apoptosis, the entire structure of the ventral nerve cord was

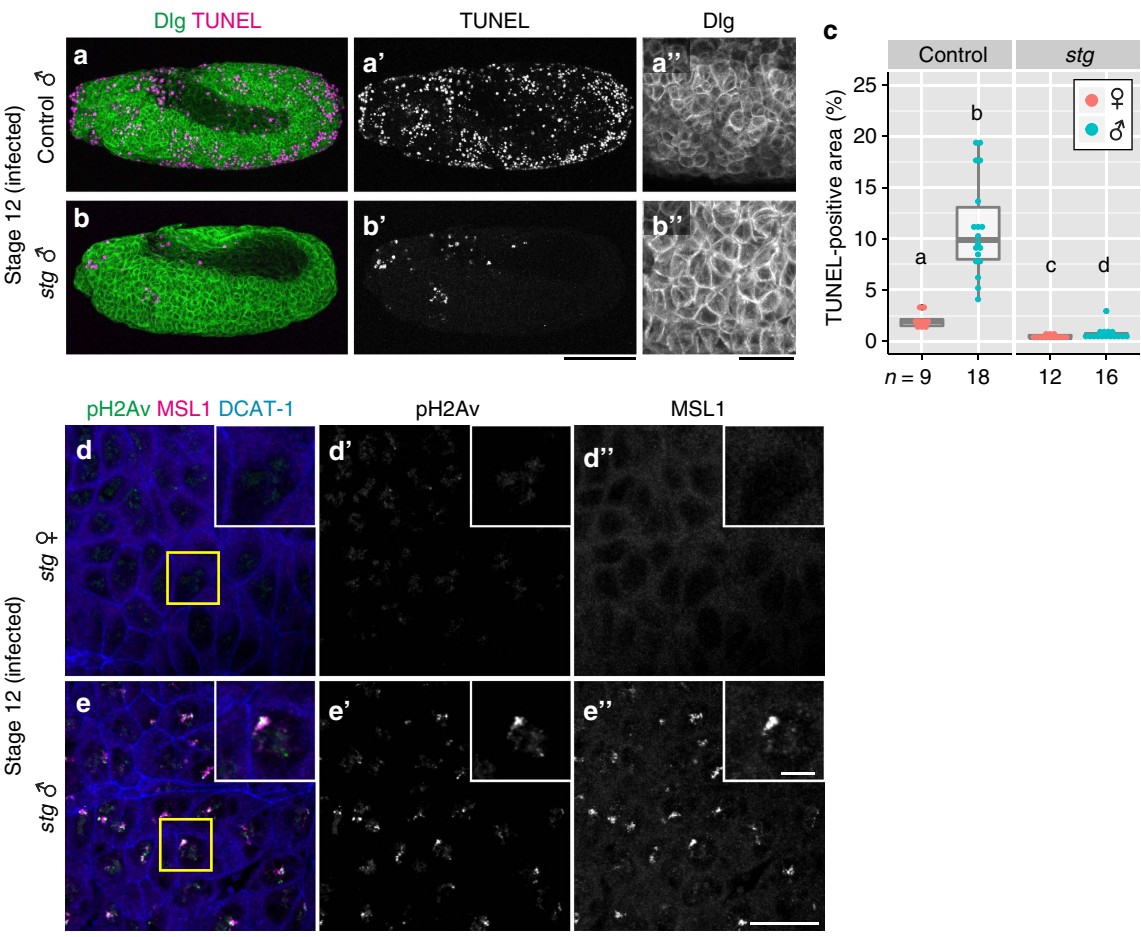

**Figure 6 | DNA damage and apoptosis in *Spiroplasma*-infected embryos mutant for *stg*.** (**a,b**) *Spiroplasma*-infected control male embryo (genotype *stg*$^{AR2}$/*TM3* or *stg*$^{7B}$/*TM3*) and *stg* mutant male embryo (genotype *stg*$^{AR2}$/*stg*$^{7B}$) at stage 12. Cell membranes and apoptotic cells are visualized by anti-Discs large (Dlg; green) and TUNEL (magenta) staining, respectively. Single-channel images of TUNEL staining and high magnification images of Dlg staining are shown in **a′–b′** and **a″–b″**, respectively. (**c**) Quantification of TUNEL-positive areas in infected control and *stg* mutant embryos. Box plots are as in Fig. 2i,j. Different letters (a–d) indicate statistically significant differences (*P* < 0.05 for *stg* females versus males, *P* < 0.01 for the others; Kruskal–Wallis test followed by Mann–Whitney *U*-tests). Sample sizes are indicated at the bottom. (**d,e**) Infected *stg* mutant female (*n* = 7) and male (*n* = 11) embryos at stage 12, in which DNA damage (pH2Av; green), the X chromosome (MSL1; magenta), and cell membrane (DCAT-1; blue) are visualized. Single-channel images of pH2Av signals and MSL1 signals are shown in **d′–e′** and **d″–e″**, respectively. Boxed regions in **d** and **e** are magnified in insets of **d–d″** and **e–e″**. Scale bars, 100 μm (**a–b′**), 25 μm (**a″,b″**), 20 μm (**d–e″**) and 5 μm (insets in **d–e″**).

considerably restored (Supplementary Fig. 5g,h), in comparison with *Spiroplasma*-infected control embryos (Supplementary Fig. 5c,d), but the structure of each neural cluster remained abnormal (Supplementary Fig. 5g,h), which was a reminiscent of the *p53* mutant phenotype (Supplementary Fig. 5e,f). These results suggest that the neural malformation in *Spiroplasma*-infected male embryos is, at least partly, a secondary effect of p53-dependent massive apoptosis, whereas the possibility that some apoptosis-independent pathway(s) may underlie *Spiroplasma*-induced neural defects cannot be excluded.

**Discussion**

In this study, we uncovered a number of previously unrecognized molecular and cellular aspects underlying *Spiroplasma*-induced male-killing during *Drosophila*'s embryogenesis, which include: (i) a large number of genes related to DNA damage and apoptosis are up-regulated specifically in *Spiroplasma*-infected male embryos (Fig. 1; Supplementary Fig. 1; Supplementary Data 1); (ii) *Spiroplasma* causes DNA damage on the male X chromosome

interacting with the functional MSL complex (Fig. 3 and Supplementary Fig. 2); (iii) the damaged male X chromosome exhibits chromosomal bridge and breakage during cell division (Fig. 4 and Supplementary Fig. 3); (iv) the functional MSL complex is not only necessary but also sufficient for triggering *Spiroplasma*-induced DNA damage, chromatin bridge and apoptosis (Fig. 5 and Supplementary Fig. 4); (v) bridge breakage in the male X chromosome is responsible for abnormal apoptosis via p53-dependent pathways (Fig. 2); and (vi) the mitosis-associated chromatin bridge-breakage is preceded by the induction of chromosome-specific DNA damage (Fig. 6). On the basis of these results, we propose a hypothetical model as to what molecular and cellular mechanisms are operating in the developmental events of *Spiroplasma*-infected male embryos, which finally result in massive apoptosis and associated developmental abnormalities leading to male-specific embryonic lethality (Fig. 7). In conclusion, *Spiroplasma* targets the dosage-compensated male X chromosome with the clue of the functional MSL complex and somehow introduces DNA damage on it, thereby causing male-specific chromosomal segregation defects

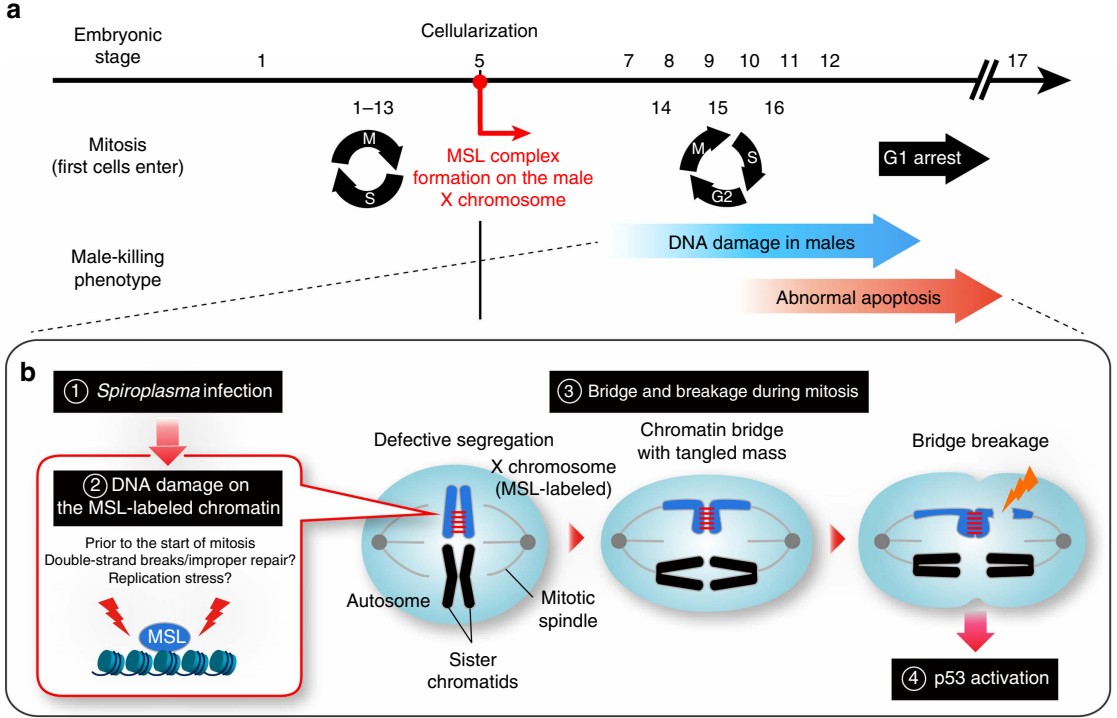

**Figure 7 | Model for the mechanism of *Spiroplasma*-induced male-killing in *Drosophila*.** (**a**) The time line of male-killing phenotypes during embryogenesis. Mitotic cycles are shown below the line. First 13 cleavage cycles are rapid and synchronous, consisting of only S and M phases. After cellularization at stage 5, cells obtain G2 phase and undergo three rounds of mitosis with specific pattern and timing (cycle 14–16), followed by G1 arrest. (**b**) The cytological model of *Spiroplasma*-induced male-killing. See the text for details.

and recruiting host's p53-dependent pathways to induce apoptosis.

It remains unknown how *Spiroplasma* damages the MSL-bound male X chromosome. On the grounds that (i) *Spiroplasma* is enriched extracellularly in *Drosophila* hosts[57], (ii) the X chromosome located within the nucleus is damaged in *Spiroplasma*-infected male embryos (this study) and (iii) mosaic and gynandromorph analyses reveal specific killing of male cells even when male cells and female cells coexist in the same embryos[26,58], it is conceivable, although speculative, that *Spiroplasma*-produced factors, so-called effectors or toxins, may be involved in the process. Some bacterial toxins, such as colibactin of *Escherichia coli*, typhoid toxin of *Salmonella typhi* and cytolethal distending toxins of various Gram-negative bacteria, are known to cause DNA crosslinking and induce double-strand breaks in eukaryotic cells, though probably not specific to sex chromosomes[59,60]. In this context, it may be notable that the *Spiroplasma* genome encodes specific prophages[61] and a plenty of phage particles are found in *Spiroplasma*-infected *Drosophila* hosts[62,63]. It has been reported that a bacterial endosymbiont *Hamiltonella defensa* produces a phage-encoded toxin, thereby protecting its aphid host against parasitoid wasps[64]. Similar symbiont-mediated defence against natural enemies has been found in several *Spiroplasma*-associated *Drosophila* species[65,66], wherein a symbiont-derived ribosome-inactivating protein was identified as a defensive factor[67]. Meanwhile, the possibility cannot be excluded that *Spiroplasma* may act on the host cells to induce some eukaryotic factors that interact with and damage the MSL-bound X chromosome. Future studies should focus on these possibilities.

Our finding that the functional MSL complex is necessary as well as sufficient for triggering *Spiroplasma*-induced male-killing (Fig. 5 and Supplementary Fig. 4) implies that the *Spiroplasma*-induced damage on the male X chromosome depends on the functional MSL complex either directly or indirectly. A simple scenario is that the protein complex itself serves as a molecular target. In this context, it may be relevant that MSL proteins evolve rapidly under strong positive selection in *Drosophila*, suggesting the possibility of evolutionary arms race between the host's dosage compensation system and the symbiont's selfish reproductive manipulation[68]. Recently, it was reported that *Spiroplasma* infection alters the localization of the MSL complex in male embryos, suggesting that *Spiroplasma* directly targets the dosage compensation machinery to induce genome-wide disruption of gene expression[24]. On the other hand, considering that the functional MSL complex is associated with various types of histone modifications and subsequent structural/transcriptional changes[5], *Spiroplasma* may influence these modifications rather than target the MSL complex itself. Recent studies revealed that some bacteria can affect chromatin structure and transcriptional activity of host cells by modulating diverse epigenetic factors such as histone modifications, DNA methylation and chromatin-associated complexes[69]. It was reported that acetylation of histone H4K16, one of the major chromatin modifications mediated by the MSL complex, weakens nucleosome packing, thereby making chromatins more accessible for DNA binding factors[70]. It is possible that the MSL-bound X chromosome similarly becomes susceptible to *Spiroplasma*-induced DNA damage.

In theory, any male-specific essential molecular, cellular and/or structural aspects of host organisms can potentially be exploited by symbiotic microorganisms to induce male-killing[9]. Probably reflecting this, symbiont-induced male-killing has evolved repeatedly in diverse bacterial lineages including *Wolbachia*, *Spiroplasma*, *Arsenophonus*, *Rickettsia* and others, where the symbiotic bacteria interact with a variety of host's molecular and

cellular components[8,9]: the male-killing *Spiroplasma* damages the dosage-compensated male X chromosome bound by the MSL complex in *Drosophila melanogaster* (this study); a male-killing *Wolbachia* suppresses host's masculinizing gene expression and thereby disturbs dosage compensation of the male Z chromosome in *Ostrinia* moths[29]; another male-killing *Wolbachia* induces defective chromatin remodelling and subsequent abnormal mitotic spindle formation in the male embryos of *Drosophila bifasciata*[16]; and a male-killing *Arsenophonus* inhibits formation of maternal centrosomes required for early male development in *Nasonia vitripennis*[14]. In the light of the diversity and commonality of male-killing mechanisms, the *Ostrinia*'s *Wolbachia* is of particular interest in comparison with the *Drosophila*'s *Spiroplasma* in that: (i) the entirely different symbiotic bacteria, *Wolbachia* ($\alpha$-Proteobacteria) and *Spiroplasma* (Mollicutes), cause similar male-killing phenotypes in the entirely different insect hosts, *Drosophila* (Diptera; male heterogametic with XY chromosomes) and *Ostrinia* (Lepidoptera; female heterogametic with ZW chromosomes); (ii) both symbiotic bacteria interact with host's dosage compensation mechanisms for inducing male-killing; (iii) however, while *Wolbachia* disturbs the dosage compensation of the male Z chromosome in *Ostrinia*, *Spiroplasma* scarcely affects the dosage compensation of the male X chromosome in *Drosophila*; and (iv) *Wolbachia*'s male-killing in *Ostrinia* is due to dosage compensation defects, whereas *Spiroplasma*'s male-killing in *Drosophila* is caused by bridge breakage of the male X chromosome and subsequent p53-mediated massive apoptosis.

In this study, we provide an integrative picture as to what mechanisms underlie *Spiroplasma*-induced male-killing, which encompass molecular, chromosomal, cellular and organismal levels. By accumulating such in-depth knowledge for different host–symbiont systems, we will be able to gain insights into the diversity and commonality of symbiont's strategies for interfering with host's sex-related cellular mechanisms, which should lead to a promising avenue for broadening the frontier of cell biology towards the realm of evolutionary biology and ecology.

## Methods

Fly stocks used in this study were obtained from the Bloomington *Drosophila* Stock Center (Indiana University), the *Drosophila* Genetic Resource Center (Kyoto Institute of Technology) and several *Drosophila* researchers. RNA-seq libraries of *Spiroplasma*-infected and uninfected embryos were constructed by TruSeq RNA Sample Preparation Kit (Illumina) and sequenced by HiSeq 2000/2500 (Illumina). Short reads were aligned to the reference genome sequence of *D. melanogaster* provided by University of California, Santa Cruz (dm3, Berkeley *Drosophila* Genome Project Release 5) (Supplementary Data 2). Of all genes annotated in the *Drosophila* genome, 8,387 genes achieved at least one CPM (counts per million reads) for at least three libraries were subjected to identification of differentially expressed genes (Supplementary Data 3). Immunofluorescence staining and other cytological procedures were as described[26], which were subjected to imaging analyses using custom R scripts with the EBImage package. Further details of the methods can be found in the Supplementary Methods.

**Data availability.** Nucleotide sequence data that support the findings of this study have been deposited in the DNA Data Bank of Japan (DDBJ: http://www.ddbj.nig.ac.jp) Sequence Read Archive with the accession numbers PRJDB4469/DRA004268/SAMD00044983-SAMD00044986 (Supplementary Data 2). All other relevant data supporting the findings of this study are included within the article and its Supplementary Information files or available on request.

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

## Acknowledgements

The *Drosophila* Genetic Resource Center at Kyoto Institute of Technology, Japan and the Bloomington Stock Center, USA, provided fly stocks. The Developmental Studies Hybridoma Bank at the University of Iowa, USA, provided antibodies. We thank Takehide Murata, John Abrams, Mitzi Kuroda, Bruce Edgar and John Jaenike for providing fly strains, John Lucchesi for providing antibodies, Yoichi Kamagata and Takafumi Mizuno for confocal microscopy, Shuji Shigenobu and Tomoko Shibata for RNA-seq data acquisition, Takashi Kiuchi and Lemaitre lab members for comments on the manuscript, and Junko Makino and Wakana Kikuchi for technical and secretarial assistance. This work was supported by Japan Society for the Promotion of Science (JSPS) KAKENHI Grant Number 12J05307 to TH. T.H. was also supported by JSPS Fellowship for Young Scientists. RNA-seq analysis was supported by KAKENHI Grant Number 22128001. Part of this work, including the efforts of T.H. and B.L., was funded by the European Research Council (ERC) Advanced Grant 339970 and the Swiss National Science Foundation (SNSF) Sinergia grant no. CRSII3_154396.

## Author contributions

T.H. and T.F. conceived the study. T.H. performed most of the experiments. H.A. carried out some of the genetic experiments. T.H, B.L and T.F wrote the paper. All authors edited and commented on the paper.

## Additional information

**Accession codes:** Nucleotide sequence data have been deposited in the DNA Data Bank of Japan (DDBJ: http://www.ddbj.nig.ac.jp) Sequence Read Archive with the accession numbers PRJDB4469/DRA004268/SAMD00044983-SAMD00044986 (Supplementary Data 2).

**Competing financial interests:** The authors declare no competing financial interests.

