## [Peer review file · Nature Communications]

Reviewers' comments:

Reviewer #2 (Remarks to the Author):

This manuscript reports the very interesting observation that Spiroplasma killing of male *Drosophila* embryos, which has been previously shown to depend on the dosage compensation complex, is associated with elevated damage to the X chromosome. The authors observe frequent X chromosome bridges in infected males, and in females that inappropriately form dosage compensation complexes, suggesting that DNA damage produces cycles of breakage that contribute to apoptosis. This idea is supported by examination of a string mutant, with reduced cell division. The genetic analysis of the basis of Spiroplasma-induced pathology, including partial suppression by loss of p53 and an apoptosis block, is convincing.

The authors also examine gene expression in infected and uninfected males and females. They find, as expected, a large amount of misregulation in infected males, but much less in infected females. In agreement with the author's contention that Spiroplasma-induced DNA damage is the basis of male-killing, up-regulated genes in infected males include indicators of DNA damage and cell death. However, the description of GO analysis in the results section (p 6-7) should be cleaned up considerably. How are the genes in cluster 4 (preferentially suppressed in uninfected males) identified, and what does this designation mean (top p 7)?

A recently published study that similarly determined the MSL complex to be necessary and sufficient for male killing, and identified elevated expression of apoptotic genes in dying males, should be cited (Curr. Bio. 10, 1339). The molecular and genetic analysis of the present study is more extensive and thoroughly done than the previous publication, although conclusions are similar.

Minor points

The statement that p53-dependent apoptosis is not responsible for neural disorder (p 14) in Spiroplasma-infected males is confusing in light of the fact that the authors observe DNA damage in neuroblast cells, and neural disorder is partially rescued by mutation of p53 and a genetic block of apoptosis.

Revise title to more accurately reflect both p53 dependent and independent phases of apoptosis. Perhaps leaving p53 out of the title would cover the necessary bases.

Explain basis of apoptosis block in H99 deletion. Is analysis done in homozygous or heterozygous H99 flies?

Briefly describe how "TUNEL positive area" is calculated (Figs. 2, 5, 6).

Some legends lack complete description of box and whisker plots.

The concluding sentence "Spiroplasma recognizes the dosage compensated male X chromosome . . ." (p 16) suggests that Spiroplasma itself is present in neural tissue. Although the following paragraph tends to dispel this idea, superficial readers may come away misinformed.

Align labels with columns on panel C of supplemental Fig. 1.

Reviewer #3 (Remarks to the Author):

This is a compelling manuscript by Harumoto, et al., investigating the mechanisms by which a specific type of symbiotic bacteria, *Spiroplasma*, causes the well-described male-specific embryonic lethality in *Drosophila melanogaster*. Several assays were used in a multiple-pronged approach including: transcriptome analyses, classical genetics, and immunofluorescent imaging to determine a spatial and temporal model of how *Spiroplasma* induce male killing in *Drosophila melanogaster*. Harumoto et al. demonstrate that many DNA repair and apoptosis genes are upregulated in male embryos infected with *Spiroplasma* (in comparison to females infected and males uninfected), suggesting that embryonic lethality is due to apoptosis and an upregulation of the DNA repair pathway. They use classical genetic approaches to demonstrate that the abnormal apoptosis is p53-dependent, as apoptosis associated with infection is delayed in p53 mutants. To determine the specific cause of the apoptosis spatially, imaging of γ H2av staining (a *Drosophila* homolog of γ H2ax, a standard marker for DNA damage and specifically DNA double-strand breaks) was used. The authors showed that γ H2av staining of embryonic chromosomes co-stained with the *msl* locus (male sex lethal, required for sex determination) on the X chromosome, suggesting that the DNA damage that occurs after *Spiroplasma* infection is specific to the X chromosome. They define the specific type of genome instability in the form of anaphase bridges and chromosomal fragmentation, which are also specific to the X chromosome, as marked by co-localization of signals of the *msl* locus. They use classical genetics to demonstrate the MSL locus is necessary for DNA damage and apoptosis in the embryo. Interestingly, they use eloquent genetics to demonstrate the MSL locus in females is sufficient to induce DNA damage and abnormal apoptosis by *Spiroplasma* infection. Lastly, the researchers use classical genetics to temporally determine when the damage occurs, concluding that *Spiroplasma* infection results in DNA damage specific to the X chromosome, which leads to anaphase bridges and gross chromosomal aberrations of the X chromosome, which then leads to p53 dependent apoptosis. This is a well-written and thorough study, which will be of interest to those in the multiple fields, including of symbiotic bacterial infection, *Drosophila*, DNA damage, and sex determination/dosage compensation. I recommend acceptance of this manuscript with a few (mostly) minor revisions/suggestions that could strengthen the manuscript:

1. Line 82, I believe they mean to reference Supplementary Fig. 1C.
2. Line 85, Figure 1E should also be referenced.
3. Line 139 and 163 (and throughout the manuscript), the authors claim that developmental apoptosis is not affected. However, it's not obvious to me the significance of developmental apoptosis, nor am I convinced of the developmental apoptosis from the images shown. Is developmental apoptosis of the embryonic head region common knowledge in the field? If not, I'd appreciate explanation of the importance of this difference. Because the authors spend time demonstrating that developmental apoptosis is not affected under conditions that impacts aberrant apoptosis, quantification may be necessary to support this claim and would strengthen their claim.
4. The authors use γ H2av staining as a marker for DNA damage, which typically is recruited to DSBs. They spend quite some time in the manuscript explaining the potential biological difference between nuclear-wide staining (apoptosis) vs. foci staining (DNA damage). Have they looked at μ 2 staining (or GFP- μ 2)? This could potentially be a way to reduce the nuclear-wide staining that the authors attribute to apoptosis, resulting in stronger conclusions.
5. Figure 1C, the X-axis labels are difficult to read and see which group is associated with each data bar. Perhaps making the text horizontal or vertical (rather than diagonal)? The same is true for Supplementary Figure 1C.
6. Figure 1F is never referenced in the text of the manuscript.

7. TUNEL staining was used throughout the study to identify apoptotic cells. However, TUNEL staining can also identify necrotic cells. A supplemental figure with Caspase-3 staining of a few of the treatments would strengthen their claim that the TUNEL staining is in fact specific for the population of cells in apoptosis.

Point-by-point response to the referees' comments

First of all, we would like to thank the reviewers for reading our manuscript carefully and providing constrictive suggestions for improvement.

[Responses to Reviewer #2]

> However, the description of GO analysis in the results section (p 6-7) should be cleaned up considerably. How are the genes in cluster 4 (preferentially suppressed in uninfected males) identified, and what does this designation mean (top p 7)?

In response to the comment, we thoroughly revised this section (line 94-122 in the revised manuscript). Specifically, we removed subheadings and re-described characteristics of each cluster so that it is easier for readers to follow the logic.

> A recently published study that similarly determined the MSL complex to be necessary and sufficient for male killing, and identified elevated expression of apoptotic genes in dying males, should be cited (*Curr. Bio.* 10, 1339). The molecular and genetic analysis of the present study is more extensive and thoroughly done than the previous publication, although conclusions are similar.

As suggested, we cited Cheng et al. (*Curr Biol.* 2016; 26(10):1339-1345; ref. 24) in the Introduction (page 4, line 57), Results (page 12, line 242) and Discussion (page 17, line 357).

> The statement that p53-dependent apoptosis is not responsible for neural disorder (p 14) in *Spiroplasma*-infected males is confusing in light of the fact that the authors observe DNA damage in neuroblast cells, and neural disorder is partially rescued by mutation of p53 and a genetic block of apoptosis.

We deleted the statement and revised the text to avoid confusion (page 14, line 289).

> Revise title to more accurately reflect both p53 dependent and independent phases of apoptosis. Perhaps leaving p53 out of the title would cover the necessary bases.

As suggested, we removed "p53-mediated" from the original title.

> Explain basis of apoptosis block in H99 deletion. Is analysis done in homozygous or heterozygous H99 flies?

We added a brief description of the *H99* deficiency strain in the main text (page 14, line 296). We also modified Supplementary Fig. 5 legends to explicitly explain the use of homozygous embryos.

> Briefly describe how "TUNEL positive area" is calculated (Figs. 2, 5, 6).

We added a more detailed explanation in the “Imaging analysis” section of Supplementary Methods.

> Some legends lack complete description of box and whisker plots.

The correction was made in Fig. 5 and 6 legends (page 34 and 35, respectively).

> The concluding sentence "Spiroplasma recognizes the dosage compensated male X chromosome . . ." (p 16) suggests that Spiroplasma itself is present in neural tissue. Although the following paragraph tends to dispel this idea, superficial readers may come away misinformed.

On account of the comment, we replaced “recognizes” by “targets” to keep consistency (page 15, line 324 and page 2, line 24)

> Align labels with columns on panel C of supplemental Fig. 1.

We modified the labels on Fig. 1c as suggested (page 29).

[Responses to Reviewer #3]

1. Line 82, I believe they mean to reference Supplementary Fig. 1C.

2. Line 85, Figure 1E should also be referenced.

Thank you very much for checking our manuscript in detail.

As for line 82 (in the original manuscript), we believe that Fig. 1c, but not Supplementary Fig. 1c corresponds to the text. We first describe general expression changes between samples (Fig. 1c), and then we focus on expression changes in each chromosome (Supplementary Fig. 1c) in the next part.

As for line 85 (in the original manuscript), we are sure that Fig. 1e (enriched GO categories) is not appropriate in this part, where we describe changes of gene expression. Fig. 1e is suitable for the next section, where we deal with the results of the GO enrichment analysis.

3. Line 139 and 163 (and throughout the manuscript), the authors claim that developmental apoptosis is not affected. However, it's not obvious to me the significance of developmental apoptosis, nor am I convinced of the developmental apoptosis from the images shown.

In response to the comment, we adjusted brightness/contrast of TUNEL images, and the edges of the embryonic epidermis were highlighted by dashed yellow lines in Fig. 2c-h to improve visibility (page 30).

Is developmental apoptosis of the embryonic head region common knowledge in the field? If not, I'd appreciate explanation of the importance of this difference. Because the authors spend time demonstrating that developmental apoptosis is not affected under conditions that impacts aberrant apoptosis, quantification may be necessary to support this claim and would strengthen their claim.

Yes. Developmental apoptosis during *Drosophila* embryogenesis was thoroughly described in 1990s (Abrams et al., *Development* 1993; 117(1): 29-43). According to this paper, the first apoptotic cells are invariably observed in the dorsal region of the head at stage 11 and become widespread in the cephalic region as shown in Fig. 2c,e,f (arrows). In the subsequent developmental stages, apoptotic cells are more prominent throughout the body (see Fig. 2a''). Furthermore, in our previous paper (Harumoto et al., *PLoS Pathog.* 2014; 10(2): e1003956), we described the progress of apoptosis in

Spiroplasma-infected embryos in detail and confirmed these observations (please see Referee Fig. 1). To clarify this point and promote better understanding, we modified the main text and cited the references above (page 7, line 134).

Referee Fig. 1 (from Harumoto et al., 2014)

4. The authors use γ H2av staining as a marker for DNA damage, which typically is recruited to DSBs. They spend quite some time in the manuscript explaining the potential biological difference between nuclear-wide staining (apoptosis) vs. foci staining (DNA damage). Have they looked at mu2 staining (or GFP-mu2)? This could potentially be a way to reduce the nuclear-wide staining that the authors attribute to apoptosis, resulting in stronger conclusions.

We have not used MU2 (Mutator 2) antibody or GFP-MU2, another DSB marker in *Drosophila*, because we believe that they show a similar pattern to γ H2Av staining. In support of this notion, it has already been demonstrated that; i) MU2 co-localizes with γ H2Av foci, and ii) it physically interacts with γ H2Av (Dronamraju and Mason, *PLoS Genet.* 2009; 5(5):e1000473). As it is well known, chromosome-wide DNA fragmentation is one of the features of apoptosis, and this fragmentation is also labeled with γ H2Av (Rogakou et al., *J Biol Chem.* 2000; 275(13):9390-9395), suggesting that any DSB markers associated with γ H2Av will detect apoptosis as nuclear-wide staining.

Meanwhile, we would appreciate this comment, because GFP-MU2 must be useful for our future study, for example, live-imaging of DSB induction during embryogenesis.

5. Figure 1C, the X-axis labels are difficult to read and see which group is associated with each data bar. Perhaps making the text horizontal or vertical (rather than diagonal)? The same is true for Supplementary Figure 1C.

As suggested, we modified the figures (see new Fig. 1c, f and Supplementary Fig. 1c).

6. Figure 1F is never referenced in the text of the manuscript.

Please note that Fig. 1f is correctly referred to in page 6, line 99 of the original manuscript (please also see the identical part of the revised manuscript).

7. TUNEL staining was used throughout the study to identify apoptotic cells. However, TUNEL staining can also identify necrotic cells. A supplemental figure with Caspase-3 staining of a few of the treatments would strengthen their claim that the TUNEL staining is in fact specific for the population of cells in apoptosis.

The reviewer is right and we agree that specificity of TUNEL staining is an important point in general. We have already addressed the issue in our previous paper (Harumoto et al., *PLoS Pathog.* 2014; 10(2): e1003956), where we carried out an intensive analysis on abnormal cell death in *Spiroplasma*-infected male embryos. In brief, i) anti-cleaved Caspase-3 antibody staining showed similar spatiotemporal patterns as those of TUNEL staining (Referee Fig. 2, left), and ii) there are no TUNEL positive cells in apoptosis-deficient *H99* mutant embryos infected with *Spiroplasma* (Referee Fig. 2, right). These data clearly demonstrate that TUNEL staining in the infected embryos specifically detect apoptosis, and not other types of cell death.

Referee Fig. 2 (from Harumoto et al., 2014)

REVIEWERS' COMMENTS:

Reviewer #3 (Remarks to the Author):

The major points of this review were covered in the first review of the manuscript. The authors appropriately and thoroughly addressed all major and minor concerns. I have nothing additional to add. I'd recommend accepting this manuscript for publication in the current revised form.